# Co–Ce Oxides Supported on SBA-15 for VOCs Oxidation

Jean-Luc Blin [1], Laure Michelin [2,3], Bénédicte Lebeau [2,3], Anton Naydenov [4], Ralitsa Velinova [4], Hristo Kolev [5], Pierrick Gaudin [1], Loïc Vidal [2,3], Anna Dotzeva [5], Krasimir Tenchev [5] and Silviya Todorova [5,*]

[1] Université de Lorraine/CNRS, L2CM, UMR7053, F-54506 Vandoeuvre-Lès-Nancy, France; jean-luc.blin@univ-lorraine.fr (J.-L.B.); pierrick.gaudin@univ-lorraine.fr (P.G.)
[2] Université de Haute Alsace (UHA), CNRS, IS2M, UMR 7361, 68093 Mulhouse, France; laure.michelin@uha.fr (L.M.); benedicte.lebeau@uha.fr (B.L.); loic.vidal@uha.fr (L.V.)
[3] Université de Strasbourg, F-6700 Strasbourg, France
[4] Institute of General and Inorganic Chemistry, Bulgarian Academy of Sciences, Acad. G. Bonchev St., Bldg. 11, 1113 Sofia, Bulgaria; naydenov@svr.igic.bas.bg (A.N.); raligeorgieva@svr.igic.bas.bg (R.V.)
[5] Institute of Catalysis, Bulgarian Academy of Sciences, Acad. G. Bonchev St., Bldg. 11, 1113 Sofia, Bulgaria; hgkolev@ic.bas.bg (H.K.); anna200485@gmail.com (A.D.); tenchev@ic.bas.bg (K.T.)
* Correspondence: todorova@ic.bas.bg; Tel.: +359-2-9792576

**Abstract:** Reported here are new data on the structural and catalytic properties of a series of mono-component cobalt and bi-component Co–Ce catalysts supported on SBA-15 (Santa Barbara Amorphous-15)). The catalysts performance has been evaluated by tests on combustion of methane, propane, and *n*-hexane. It was established that the preparation of the Co–Ce catalysts by the 'two-solvent' technique does not significantly change the mesoporous structure, however, its pores are clogging with the Co and Ce guest species. Cobalt and cerium are uniformly distributed and preferentially fill up the channels of SBA-15, but oxide agglomerates located on the surface are observed as well. The highest activity of the mono-component cobalt sample is explained by its higher reducibility as a result of lower interaction of the cobalt oxide with the SBA-15. The fine dispersion of cobalt and cerium oxide and their strong interaction in the channels of the SBA-15 molecular sieve, leads to the formation of difficult-to-reduce oxide phases and, consequently, to lower catalytic activity compared to monocomponent cobalt oxide catalyst. The synthesised mesoporous structure can prevent the agglomeration of the oxide particles, thus leading to the successful development of a new and stable catalyst for decreasing greenhouse gas emissions.

**Keywords:** SBA-15; Co–Ce oxides; catalytic combustion; methane; propane; *n*-hexane oxidation; mesoporous structure

## 1. Introduction

The major air pollutants are volatile organic compounds (VOCs). They contribute to a number of environmental problems such as formation of ground-level ozone, formation of photochemical smog, and toxic air emissions. The European Commission presented the so-called Green Deal in December 2019, also known as the Green Pact, which is a set of policies aimed at making Europe climate neutral by 2050. The plan is to reduce greenhouse gas production at least by 50% compared to 1990 levels, as well as zero pollution by 2050, whether soil, air, or water pollution. There are many different techniques for VOCs removal, such as adsorption, absorption, biofiltration, combustion (incineration), and catalytic oxidation. The choice of the technique to be used depends on the VOCs nature and concentration and waste gas flow rate. Catalytic incineration is a competitive abatement technique for VOCs' neutralization, especially when the organics cannot be recycled, sold, or are present in low concentration. Compared with thermal incineration, catalytic oxidation occurs at lower temperatures and, thus, it requires lower energy cost and gives no NOx emission. Metal oxides or supported noble metals are used in catalytic oxidation. Metal oxides are alternative to noble metals as catalysts for total oxidation [1].

Cobalt oxide is reported to be quite promising among the metal oxides used for preparation of supported catalysts for the removal of CO, even at room and lower temperatures [2], but is a toxic element [3]. Wang et al. [4] determined the key factors for the extraordinarily high catalytic activity of $Co_3O_4$-based catalysts: CO adsorption strength, the barrier of CO reacting with the lattice oxygen, and the redox properties of $Co_3O_4$. The combination of $Co_3O_4$ and other oxides such as $CeO_2$, $ZrO_2$, $Al_2O_3$, etc. often affects the textural, morphological, redox, acidic, and basic properties, which lead to increase the catalytic activity. Cerium oxide has been widely used in the automotive three-way catalysts as an oxygen storage medium and thermal stabilizer. In this context, the oxygen storage capacity of cerium oxide is associated with the fast $Ce^{4+}/Ce^{3+}$ redox process, making more oxygen available for the oxidation process. On the other hand, ceria can enhance the dispersion of supported metals or stabilize the oxide support against thermal sintering [4,5].

A very significant disadvantage of oxide catalysts for the oxidation of volatile organic compounds is the sintering of the active phase during the reaction. Therefore, various methods or carriers with a highly developed surface are used for catalysts preparation in order to stabilize the oxide particles. Studies have shown that the porosity of support modifies the catalytic properties by affecting the particle dispersion and reducibility of metal species [6].

SBA-15 (Santa Barbara Amorphous-15) is the most used carrier among the mesoporous silicas because it possesses a regular hexagonal array of pores with uniform diameter, a very high specific surface area and high pore volume, it is inert and stable at elevated temperature, and has good mechanical stability.

In our previous papers [6], we found out that the Co–Mn catalyst supported on SBA-15 resistance towards agglomeration can be attributed to the mesoporous structure. In our earlier investigations, it was established that cerium modified the catalytic behavior of cobalt and manganese in *n*-hexane [7,8], ethyl acetate [8], and CO oxidation. The catalytic properties in these reactions depended on the sequence of introduction of active components. The introduction of ceria leads to the increase in the cobalt oxide dispersion and changes the reducibility of $Co_3O_4$. Because of close interaction between $Co_3O_4$ and $CeO_2$ in the catalyst prepared with a common solution of Co and Ce nitrates, more surface oxygen species are provided to the cobalt oxide [7]. It is, therefore, reasonable to assume that the combination of cobalt with ceria-containing SBA-15 will lead to good catalytic performance.

The present paper is focused on of mixed Co–Ce oxide catalysts for the combustion of different VOCs. The work is focused on the investigation of the structural and catalytic properties of series of mono-component cobalt and bi-component Co–Ce catalysts supported on SBA-15. The selected reagents are methane, propane, and *n*-hexane. The specified volatile organic compounds are chosen because methane is the second most abundant greenhouse gas with a global warming potential ca. 20 times greater than $CO_2$, propane is the main component in the liquefied petroleum gas (LPG) (composed of primarily propane and butane) and n-hexane, because it is the component of many products related to industry and in the air it participates in a radical reaction yielding 2-hexanone, 2-and 3-hexyl nitrate, and 5-hydroxy-2-pentanone, all of them existing in photochemical smog.

## 2. Results and Discussion

### 2.1. Characterization of the Mesoporous Catalysts before and after Reaction

The catalysts are characterized after complete of *n*-hexane oxidation, due to the fact that all tested samples show the highest activity in this reaction.

The pure SBA-15 was characterized in our previous paper [6]. Three lines at 10.7, 6.2, and 5.3 nm are visible in the Small angle X-Ray scattering (SAXS) pattern of SBA-15 (Figure 1) and they are characteristic of the (10), (11), and (20) reflections of the 2D-hexagonal arrangement of the mesoporous channels.

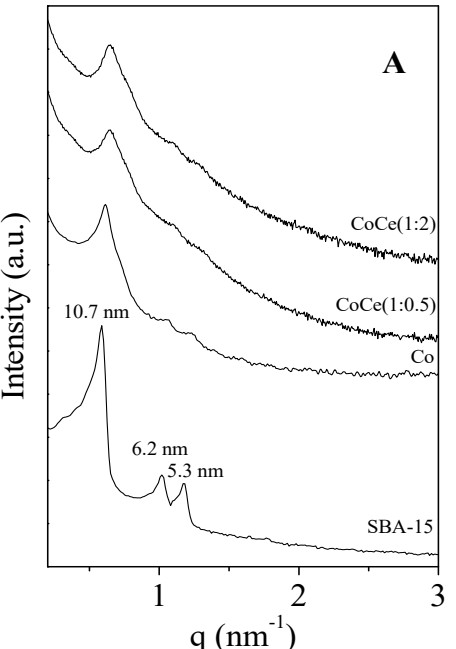 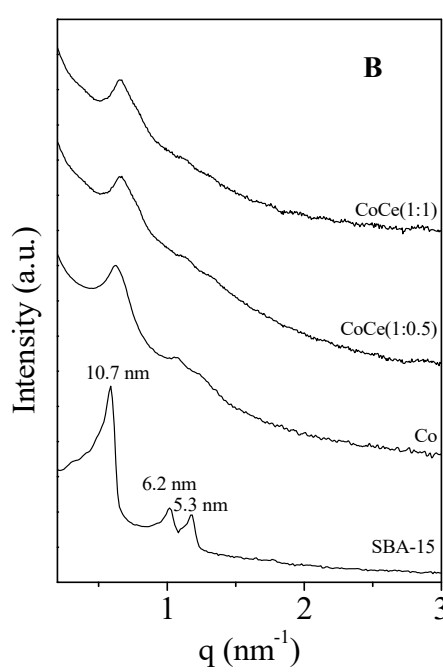

**Figure 1.** Small angle X-Ray scattering (SAXS) patterns of the catalysts before (**A**) and after (**B**) *n*-hexane oxidation.

After the addition of Co, except for a decrease in intensity of the secondary reflections, no significant change is noted in the SAXS patterns. This phenomenon is more pronounced with the further addition of cerium (Figure 1). This can suggest a mesopore filling by the Co and Ce species. After reaction, SAXS patterns present the same features than before reaction. Therefore, there is no damage of the mesopore ordering during the *n*-hexane oxidation.

The bare SBA-15 exhibits a mixed type I/type IV isotherm according to the International Union of Pure and Applied Chemistry (IUPAC) classification [9], which indicates the presence of both micropores and mesopores as expected for such ordered mesoporous silica [10]. An H1 type of hysteresis loop, in which adsorption and desorption branches are steep, has been observed and agrees with the presence of cylindrical mesopores. The specific surface area $S_{BET}$ and total pore volume $V_p$ values are 1020 m$^2$/g and 1.19 cm$^3$/g, respectively (Table 1). The mesopore diameter distribution determined by using the Barret–Joyner–Halenda (BJH) method with Kruk-Jaroniec-Sayari (KJS) correction [11] is quite narrow and it is centred at ca. 9.0 nm (Table 1). With the addition of Co and Ce the total pore volume of adsorbed nitrogen decreases, but the isotherms remain characteristic of mesoporous materials (Figure 2). In addition, the shape of the desorption branch is modified and corresponds to H5 hysteresis that indicates partial mesopores being clogged by the guest species.

Upon the addition of cobalt, the $S_{BET}$ and the $V_p$ decrease from 1020 to 454 m$^2$/g and from 1.19 to 0.52 cm$^3$/g, respectively (Table 1). The mesopore diameter decreases and a second component appears in the mesopore size distribution (Table 1). $S_{BET}$ and $V_p$ further drops to 347 m$^2$/g and to 0.31 cm$^3$/g, respectively, with the immobilization of cerium. Mesopore sizes are also slightly reduced. These variations support the partial mesopore clogging by the Co and Ce species and indicate that they are located inside the mesopores. It is noteworthy that the decreases in both $S_{BET}$ and $V_p$ are also a consequence of the partial loss of microporosity. Micropores can be strongly impacted by the impregnation process that involved hydration drying and thermal treatment at 500 °C, which could favour surface silanol condensation. They can be also partially obscured by the active phase. After the n-hexane oxidation, taking into account the uncertainty on the measurements (≈5%), no variation of the shape of the isotherm (Figure 2B), of the specific surface area, of the total pore volume and of the mesopore diameter (Table 1) is noted for mixed CoCe catalysts.

Thus, the textural properties of the mixed CoCe catalysts are preserved during the reaction. Concerning the Co catalyst, a drop of $V_p$ that is due to a decrease of both the $V_\mu$ and $V_{mes}$ is observed. This textural modification seems related to a modification of the active phase during the n-hexane oxidation reaction that could indicate that the presence of Ce allows it to stabilize the catalyst upon the oxidation reaction.

**Table 1.** D-spacing values, specific surface area ($S_{BET}$), micropore surface area ($S_\mu$), total pore volume ($V_p$), micropore volume $V_\mu$, mesopore volume $V_{mes}$, and pore diameter (∅) of the catalysts before (BR) and after (AR) reaction (*n*-hexane oxidation).

| Sample | | D-Spacing (nm) | $S_{BET}$ (m²/g) | $S_\mu$ [a] (m²/g) | $V_p$ [b] (cm³/g) | $V_\mu$ [a] (cm³/g) | $V_{mes}$ [c] (cm³/g) | ∅ [d] (nm) |
|---|---|---|---|---|---|---|---|---|
| SBA-15 | | 10.7 | 1020 | 251 | 1.19 | 0.09 | 1.10 | 10 (8) [6] |
| Co-SBA-15 | BR | 10.1 | 454 | 114 | 0.52 | 0.04 | 0.48 | 8.5−10.2 (8.7) [6] |
| | AR | 10.0 | 401 | 93 | 0.41 | 0.04 | 0.37 | 8.1 (7.3) |
| Co:Ce(1:0.5) | BR | 10.0 | 347 | 28 | 0.31 | 0.01 | 0.30 | 5.3−7.4 (6.4) |
| | AR | 10.0 | 367 | 37 | 0.31 | 0.01 | 0.30 | 5.5−7.4 (6.1) |
| Co:Ce (1:2) | BR | 9.9 | 362 | 29 | 0.32 | 0.01 | 0.31 | 5.1−7.9 (5.6) |
| | AR | 10.0 | 376 | 44 | 0.31 | 0.01 | 0.30 | 5.1−7.4 (6.1) |

[a] Micropore Surface area $S_\mu$ and volume $V_\mu$ determined from t-plot method, [b] Total pore volume $V_p$ determined at $p/p° = 0.95$, [c] Mesopore volume $V_{mes} = V_p − V_\mu$, [d] Average pore diameter obtained from the BJH method with KJS correction applied to the adsorption branch of the isotherm (values in brackets were determined from desorption branch).

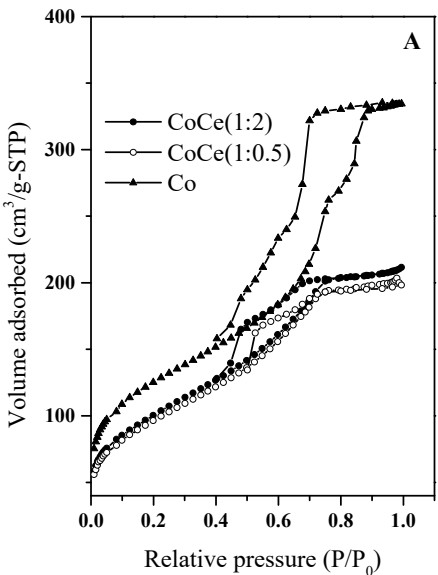
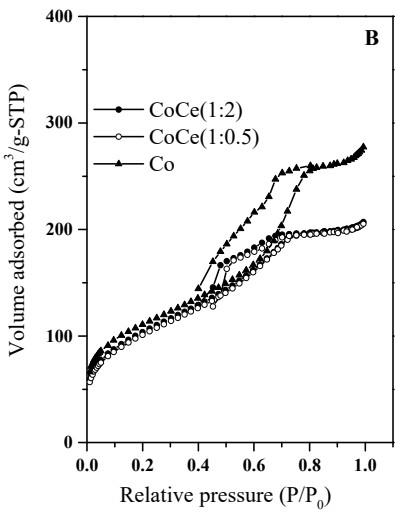

**Figure 2.** Nitrogen adsorption–desorption isotherms of the catalysts before (**A**) and after (**B**) *n*-hexane oxidation.

From the SAXS and the nitrogen adsorption–desorption analyses it can be inferred that cobalt and cerium are located in the mesopores of SBA-15.

The TEM images of the CoCe catalysts before and after reaction are illustrated in Figures 3 and 4, Figures S1 and S2 from Supplementary Materials. The clusters of $Co_3O_4$ are mostly situated on the surface and are partially located inside the pore channels of SBA-15 [6].

The TEM images show well-ordered 2D-hexagonal *p6mm* pore organization of SBA-15. The ordered pore channels were well-preserved both after the incorporation of Co and Co–Ce and after the *n*-hexane oxidation reaction. The darker areas correspond to the crystalline oxides. The Energy Dispersive X-ray (EDX) analyses were performed with all studied samples and Co and Ce element mapping (Figures S1–S3 from Supplementary Materials) confirmed uniform distribution in the CoCe samples (Table 2). From TEM images the presence of particles that fit the mesopore size suggesting that they are inside the channels for catalyst before reaction (Figure 3B,C). Dark nanowires along pores indicate that the

mesopores are filled up with oxides (Figure 3C and Figure S2 from Supplementary Materials). TEM images after reaction show that particles are bigger and/or agglomerated on the surface of the SBA-15 support (Figure 3D,E and Figure S3 from Supplementary Materials).

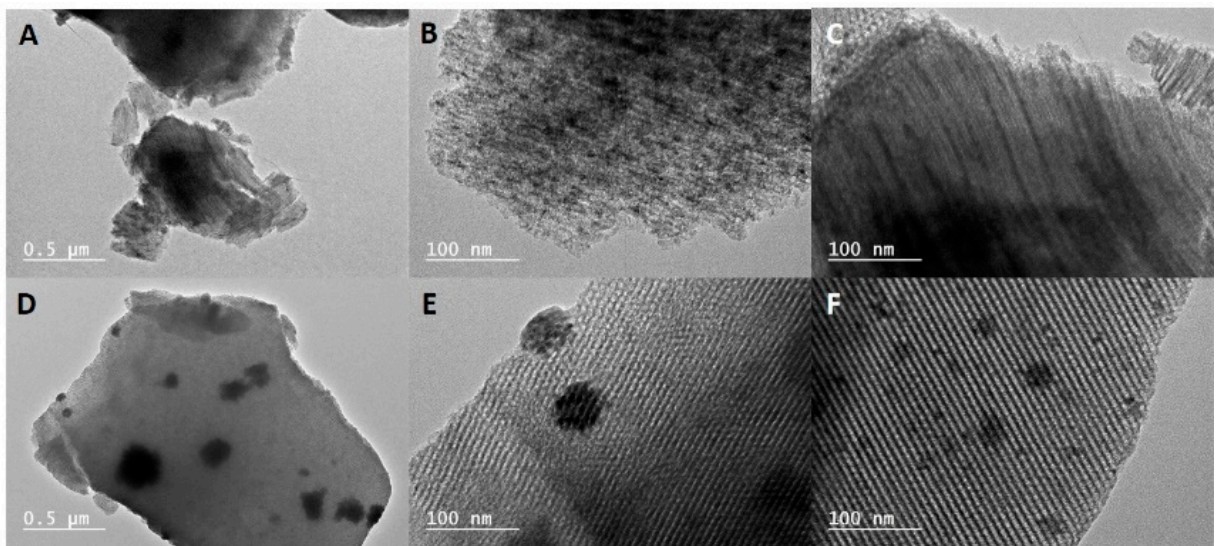

**Figure 3.** TEM images of Co:Ce (1:0.5) catalyst before (**A**–**C**) and after (**D**–**F**) reaction.

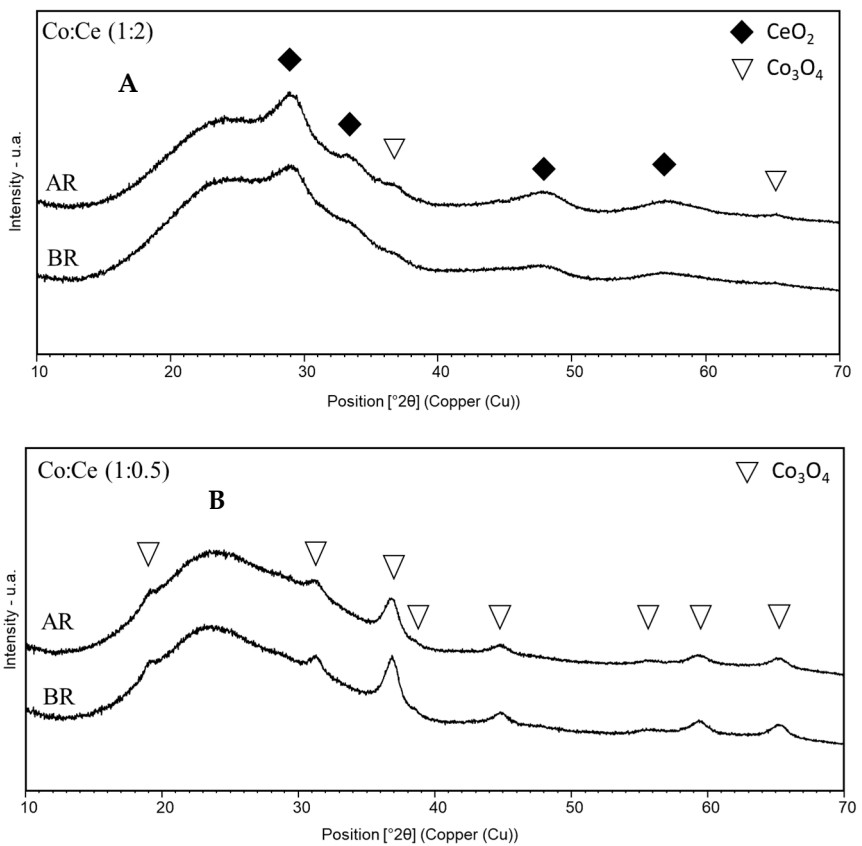

**Figure 4.** Wide angles X-ray diffraction patterns of the Co:Ce (1:2) catalyst before and after reaction (**A**) and Co:Ce (1:0.5) catalyst before and after reaction (**B**).

**Table 2.** Surface atomic concentrations (at. %) of $Co^{2+}$, $Co^{+3}$, $Ce^{+3}$, and $Ce^{+4}$.

| Sample\Element | O 1s | Si 2s | Co 2p1/2 | Ce 3d | $Ce^{3+}/Ce^{4+}$ |
|---|---|---|---|---|---|
| Ce/SBA-15 | 64.07% | 35.04% | - | 0.88% | 45/55 |
| Co/SBA-15 | | | 1.22 ($Co^{3+}$) | | |
| Co:Ce (1:0.5) | 63.32% | 34.95% | 1.37% | 0.37% | $Ce^{3+}$-100% |
| Co:Ce (1:2) | 64.04% | 34.03% | 1.49% | 0.45% | 55/45 |

The EDX analyses were performed with all studied samples and Co and Ce element mapping confirmed uniform distribution in the CoCe samples. From EDX analyses, the atomic % of Co, Ce, and Si were determined and the corresponding Co/Ce atomic ratios show that Ce was partially incorporated in the cobalt oxide.

*2.2. Characterization of the Supported Metal Species before and after Reaction*

The XRD patterns of Co:Ce (1:2) manifested broad and low intensive peaks for $CeO_2$ (International Centre for Diffraction Data (ICDD) card n°01-083-4917) and some traces of $Co_3O_4$ (ICDD card n°04-006-3982). In the case of mono component Co-SBA-15, the wide-angle XRD pattern of the Co-SBA-15 material presents low and broad peaks corresponding to crystalline $Co_3O_4$ phase [6]. Obviously, the deposition of cobalt and cerium from a common solution led to a loss in crystallinity of cobalt oxide.

The situation is slightly different for the sample Co:Ce (1:0.5) with less cerium oxide (Figure 4B). The main lines in the X-ray spectrum of this sample refer to $Co_3O_4$. No cerium oxide lines were observed, indicating fine dispersion of $CeO_2$.

Whatever the Ce content, the XRD patterns (Figure 4A,B) of bi-component CoCe-SBA-15 materials after the catalytic reaction are similar to those before reaction indicating that crystalline phases are not affected.

The oxidation states of Co and Ce on the surface is examined by X-ray photoelectron spectroscopy (XPS). Figure 5 shows an example curve fitting of $Co2p_{1/2}$ (left-hand side) and Ce3d (right-hand side) core levels of investigated samples before the catalytic activity test. $Co_{3/2}$ is more intensive in comparison to $Co_{1/2}$, but because of its overlapping with a part with Auger CoLMM peak it is reasonable to study $Co_{1/2}$ peak. The information about oxidation state of cobalt is coded in the peak binding energy together with the line form and existence of peak satellite. A peak centered at 798 eV is visible for cobalt samples modified with Ce and it is ascribed to $Co^{2+}$ for both samples marked with Co:Ce(1:0.5) and Co:Ce(1:2). The presence of $Co^{2+}$ ions in is confirmed by the relatively intensive 3d → 4s "shake-up" satellite with binding energy of 803–804 eV (Figure 6). $Co^{3+}$ ions are only surface species observed in mono-component cobalt catalyst (see Table 2) indicating that presence of Ce decreases cobalt oxidation state to $2^+$. The formation of $Co^{2+}$ in the ceria modified samples could be explained with transfer electrons to $Co^{3+}$ from $CeO_2$ because of the close contact of them [12]. As is visible from EDX Co and Ce element mappings, both elements are uniformly distributed, and it is possible to suggest intimate contact between them in the channels of SBA-15 (Figures S1–S3 from Supplementary Materials).

The variation of the oxidation state of cerium ions on the surface of the Co–Ce samples manifests itself in their X-ray photoelectron Ce3d spectra. The energy positions of the Ce3d peaks and the presence of a peak at about 916.8 eV binding energy correspond to $Ce^{4+}$ in $CeO_2$ [13]. As shown in Figure 5 (right-hand side), ten sub-peaks were used for the curve fitting procedure of the Ce3d spectra for samples marked with Co:Ce(1:2) and Ce monocomponent sample. Four peaks at about 882, 884, 899, and 903 eV corresponding to $Ce^{3+}$ and the other six peaks at about 886.8, 887.8V, 900.2 (double peak), 906.3, and 916.8 eV depicting $Ce^{4+}$ oxidation state [14]. The ratio between fitted peak areas of $Ce^{3+}$ and $Ce^{4+}$ is shown in Table 2. The sample marked with Co:Ce(1:0.5) shows only $Ce^{3+}$ oxidation state indicating that the decrease in Ce concentration stabilizes $Ce^{3+}$ on the surface.

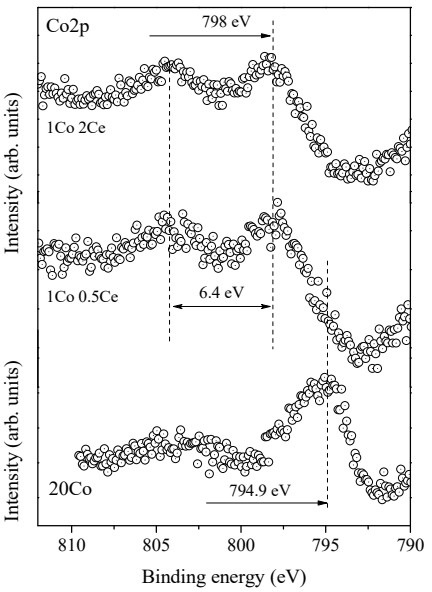
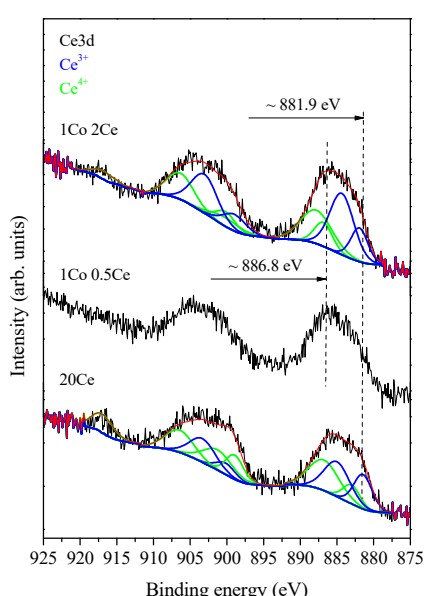

**Figure 5.** Fitted Co2p1/2 and Ce3d photoelectron peaks of the Co, Ce, and Co–Ce samples.

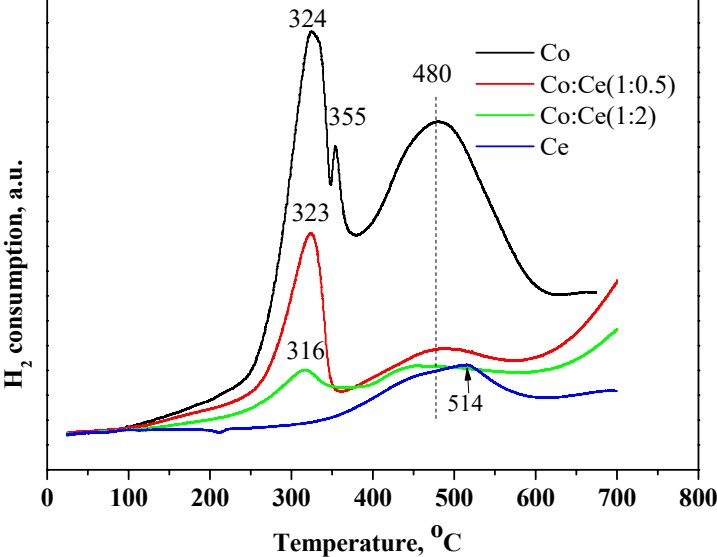

**Figure 6.** Temperature-programmed reduction (TPR) spectra of single- and bi-component catalysts.

According to literature [15,16], $Ce^{4+}$ and $Ce^{3+}$ are present together, although the kind of cerium salt used as precursor and the treatment atmosphere can affect the relative concentration.

The $O_2$-TPD, $H_2$/ TPR (Temperature-programmed reduction,) and $C_3H_8$/TPR experiments were performed in order to obtain information regarding the adsorptive and reductive properties of the catalyst. The selected temperature range is close to the one used for the catalytic activity tests; the heating temperature was limited to 450 °C for $O_2$-TPD and $C_3H_8$/TPR and 700 °C for $H_2$/TPR.

The TPR profiles of all samples are shown in Figure 6. As was described in our previous work [6], the reduction peaks at 324 and 355 °C are attributed to the reduction of large size supported $Co_3O_4$ particles and the third one, centered at 480 °C is attributed to small particles strongly interacting with the support. The main reduction feature for all bi-component samples is the presence of the reduction tail above 600 °C. The reduction peak for large size $Co_3O_4$ particles shifts to the lower reduction temperatures and this shift

is more pronounced for the Co:Ce(1:2) catalyst which could be explained with the particle size effect-the smaller the particle size the lower is the reduction temperature registered. As was shown from XRD data of this sample, no reflection patterns were observed for $Co_3O_4$ phase. The reduction above 600 °C can be explained by the presence of difficult-to-reduce phases. According to Carrero et al. [17] the reduction peak above 500 °C can be assigned to cobalt oxide species in intimate contact with the SBA-15 support. According to Huang and co-authors [18] both $Co^{2+}$ strongly interacting with $CeO_2$ and bulk oxygen of $CeO_2$ can be reduced in around 600 °C. Some authors attribute the high-temperature reduction peaks in the TPR profiles of cobalt-modified mesoporous materials to the reduction of silicates formed by the interaction of highly dispersed CoO phases with the support [19,20]. In the case of cobalt deposition by the "two-solvent" method, there is no clear evidence of silicate phase formation. It is very likely that in our case the consumption of hydrogen over 600 °C is due on the one hand to the reduction of $Co^{2+}$ ($Co^{2+}$ is confirmed from XPS data) strongly interacting with cerium oxide and, on the other hand, to the reduction of cobalt oxide strongly interacting with the framework of SBA-15. As shown by the TEM images, a large part of the cobalt oxide is finely dispersed and located inside the channels of the mesoporous carrier.

The information about the type of oxygen species on the surface was obtained by $O_2$-TPD experiment. As can be seen from the Figure 7, three desorption intervals can be distinguished in the $O_2$-TPD spectra: 20–220 °C, 220–350 °C, and 350–500 °C. The data in the current literature ascribed the low temperature desorption peak (<200 °C) to the desorption of physically adsorbed oxygen. The desorption in the interval from 200 to 400 °C is attributed to the chemisorbed oxygen, while the desorption at higher temperatures (>400 °C) corresponds to the bulk lattice oxygen in the structure [21,22]. According to Song et al. [23] the desorption within the 150−250 °C interval is ascribed to surface adsorbed peroxy $O_2^-$(ad) species, the desorption within the 280−340 °C interval— to surface adsorbed $O^-$(ad) species; desorption within the interval 350−670 °C to surface lattice oxygen $O^{2-}$ (ad/lattice) and beyond 700 °C—the desorption of bulk lattice oxygen.

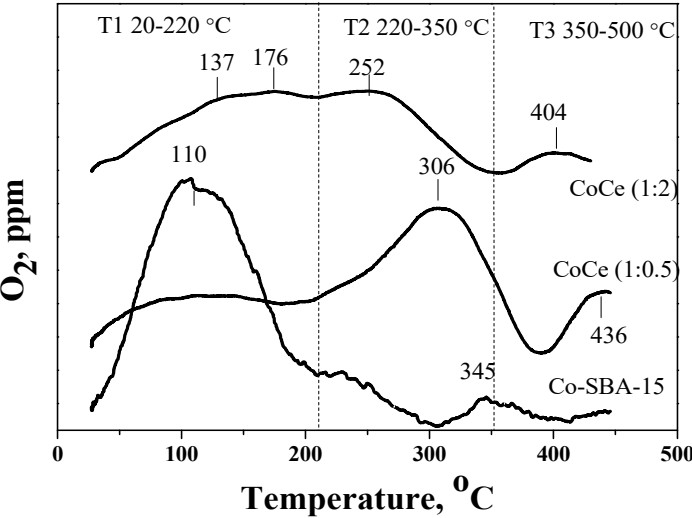

**Figure 7.** $O_2$-TPD profiles of single- and bi-component catalysts.

According to the literature data the desorption in the T1 region can be attributed to the $O_2^-$(ad), species in the T2 to the $O^-$(ad) and in the species in T3 interval to the surface lattice oxygen $O^{2-}$ (ad/lattice). As seen in Figure 7, the doping of cobalt oxide with cerium shifts desorption peaks in the intervals T2 and T3 to the lower temperature and it increases the quantity of desorbed oxygen species.

As shown in Figure 8, the complete propane oxidation in absence of gas-phase oxygen for a Co:Ce (1:0.5) sample starts at approximately 100 °C, which is evidenced by the

formation of $CO_2$, produced by the reaction of propane gas phase and the oxygen from the catalyst. The $CO_2$ formation curve consists of three main peaks—a low temperature one (centred at 114 °C) and two peaks at high temperatures—the broad one at about 287 °C and a tail above 400 °C. The low temperature peak is absent in the TPR-propane spectra of the mono-component cobalt catalyst and the samples with higher Ce concentration—Co:Ce (1:2) and the higher temperature reduction peaks move to the low temperatures. For comparison, TPR-propane spectra of the mono-component cobalt catalyst shows a single broad peak in the temperature range of 230–330 °C.

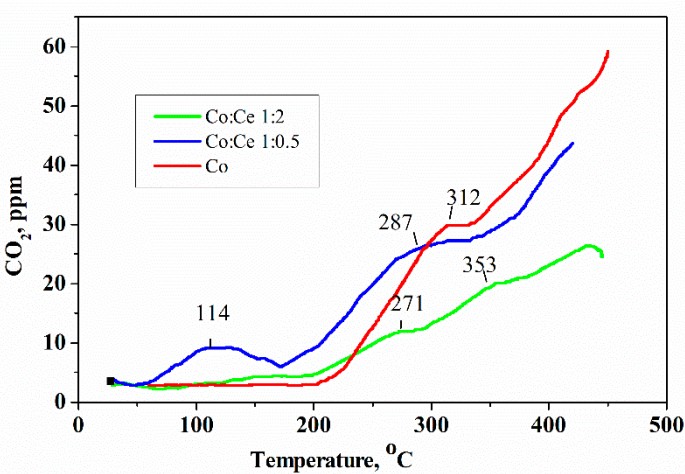

**Figure 8.** $C_3H_8$/TPR experiments of monocomponent Co and bi-component CoCe catalysts.

Considering that there are several of oxygen species on the catalytic surface, we could attribute the first low-temperature peak to the interaction of propane with the $O_2^-$(ad) species, the peaks in the interval 200–380 °C to the reduction of $O^-$(ad) species, and the peaks at >400 °C most probably are caused by the surface lattice oxygen $O^{2-}$.

### 2.3. Catalytic Activity

The temperature dependencies of the methane, propane, and n-hexane complete oxidation over the mono- and bi-component catalysts are shown in Figure 9. The only reaction products on all investigated samples were $H_2O$ and $CO_2$. The incomplete oxidation to CO occurs to a lower extent (Figure S4, Supplementary Material).

The most active catalysts in all studied reactions are mono-component cobalt. The addition of cerium oxide worsens the catalytic activity in all three reactions studied.

According to the literature data, the different parameters, such as the crystallite size, the reducibility of cobalt species, the type of support, the cobalt content, control the catalytic activity of the cobalt oxide-based catalysts for propane complete oxidation [24–27]. The interaction of the cobalt oxide species with the support is also very important because if the particle–support interaction is stronger, their reducibility decreases and as a consequence the activity drops down [28,29]. The main factors governing the catalytic activity of $CH_4$ and VOCs complete oxidation over $Co_3O_4$ and $Co_3O_4$–$CeO_2$ binary oxides are high bulk oxygen mobility (through the Mars–van Krevelen mechanism) and formation of highly active oxygen [4]. Luo et al. [30] found that the surface lattice oxygen species are related to $C_3H_8$ oxidation and, the propane oxidation takes place on neighboring surface lattice oxygen sites in $Co_3O_4$ or crystallites where propane molecule is activated.

In our previous investigations concerning bi-component CoMn samples, supported on $SiO_2$, SBA-15, and macro-mesoporous materials [6–8,31,32] we established that the *n*-hexane oxidation reaction proceeds through the Mars–van Krevelen mechanism. Thus, the catalytic behavior can be correlated with the lattice oxygen mobility and with the corresponding the catalyst reducibility

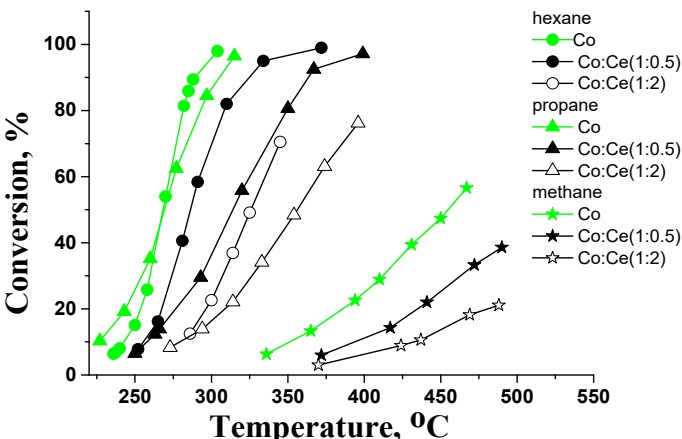

**Figure 9.** Temperature dependence of methane, propane, and *n*-hexane combustion.

The fact that the mono-component cobalt sample has the highest catalytic activity in all studied reactions can be attributed on its higher reducibility as a result of lower interaction of the cobalt oxide with the SBA-15. As was reported in our earlier work [6], the cobalt oxide in this sample forms two types of particles: large ones, $Co_3O_4$ situated on the support and small ones, strongly interacting with the support.

As shown by the TEM images and the reduction profiles, in two-component CoCe catalysts, the oxide particles are located in the channels of the mesoporous structure and there is a strong interaction between cobalt oxide and cerium oxide on the one hand and on the other, between cobalt oxide and the support, which decreases the reducibility of the samples and as a consequence their activity decreases. Another reason for low activity could be fact that in bi-component catalysts the oxide particles are situated inside the channels and are less accessible to the reagents. We observed a similar phenomenon with Co–Mn catalysts obtained by the same "two-solvent" method [6]. Obviously, this method of preparation is responsible for the localization of the oxide particles in the channels of mesoporous oxide. Despite the lower activity of the two-component Co–Ce catalysts compared to the monocomponent cobalt one, it can be supposed that they would be promising catalysts for the VOCs oxidation because they have reduced cobalt content. Cobalt is a highly toxic element [2] and its partial replacement with another element is a promising path in the development of catalysts for the VOCs oxidation.

The catalysts were examined by different physicochemical methods after the reaction of *n*-hexane oxidation and it was established that insignificant changes occur in the mesoporous structure and morphology, oxide particle size, and oxide phases. The mesoporous structure probably prevents the agglomeration of the oxide particles after the reaction.

## 3. Materials and Methods

### 3.1. Catalysts Preparation

The SBA-15 was synthesized by a sol-gel route according to a standard procedure described in [6].

Both single component cobalt and bi-component samples were prepared according to the "two-solvent" method [33]. The total metal content (Co+Ce) is about 20 wt.% and it is calculated to be supplied by the starting solutions and the Co:Ce ratio is 1:0.5 and 1:2, respectively. The samples were denoted as Co-SBA-15, Co:Ce(1:0.5), and Co:Ce(1:2), where the number represents the Ce and Co number of moles. The SBA-15 material was suspended in dry hexane (Fluka, assay >95%), used as hydrophobic solvent. Then, a desired amount of metal nitrate (Co(NO$_3$)2·6H$_2$O and Ce(NO$_3$)$_3$·6H$_2$O from Sigma-Aldrich, assay 99.99% trace metals basis) was dissolved in distilled water, the quantity corresponding to the pore volume of SBA-15 determined by N$_2$ adsorption. This aqueous solution containing metal precursors was then added dropwise to the suspension. The

solid phase was recovered by filtration and dried up in air and then calcined for 3 h at 500 °C in air atmosphere.

*3.2. Catalyst Characterization*

The obtained materials were characterized by SAXS, $N_2$ adsorption–desorption, X-ray diffraction, and TEM.

Small angle X-Ray scattering (SAXS) data were collected on a "SAXSess mc$^2$" instrument (Anton Paar, Graz, Austria), using a line-collimation system. This instrument is attached to an ID 3003 laboratory X-Ray generator (General Electric, Ahremburg, Germany) equipped with a sealed X-Ray tube (PANalytical, λ Cu, Kα = 0.1542 nm) operating at 40 kV and 50 mA. Each sample was introduced into a "Special Glass" capillary for liquids and liquid crystals (Φ = 1.5 and 2.0 mm for micellar solutions and liquid crystals, respectively), or between two sheets of Kapton® for materials, then placed inside an evacuated sample chamber, and exposed to X-Ray beam. Scattering of X-Ray beam was registered by a CCD detector (Princeton Instruments, Trenton, NJ, USA 2084 × 2084 pixels array with a 24 × 24 μm$^2$ pixel size) at 309 mm distance from the sample. Using SAXSQuant software (Anton Paar), the 2D image was integrated into one-dimensional scattering intensities I(q) as a of the magnitude of the scattering vector q = (4π/λ) sin(θ), where 2θ is the total scattering function angle. Thanks to a translucent beam-stop allowing the measurement of an attenuated primary beam at q = 0, all measured intensities can therefore be calibrated by normalizing the attenuated primary intensity. Data were then corrected for the background scattering from the cell and for slit-smearing effects by a desmearing procedure from SAXSQuant software, using the Lake method.

Nitrogen adsorption–desorption isotherms were carried out at −196 °C within a wide relative pressure (P/P$_0$) range varying from 0.010 to 0.995 with a volumetric adsorption analyzer TRISTAR 3000 manufactured by Micromeritics. Before measurements, the samples were outgassed under vacuum (pressure = 0.13 mBar) at 25 °C for 16 h. The specific surface area of each sample was calculated by the Brunauer–Emmett–Teller (BET) method [34]. The pore diameter and the pore size distribution were determined from the adsorption branch of the corresponding isotherm using the Barret–Joyner–Halenda (BJH) method [35].

Wide angles X-Ray diffraction patterns were collected on a BRUKER D8 ADVANCE A25 operating with Cu Kα radiation (Kα = 0.154,18 nm) and equipped with the LynxEye XE-T high resolution energy dispersive 1-D detector. The XRD powder patterns were collected at 25 °C within the range 3 < 2θ < 70, step = 0.018°2θ. The phases identification has been realized with the X'PertHighscore software (PANalytical) and the PDF-4+ 2020 database from the International Centre for Diffraction Data (ICDD). Transmission Electron Microscopy (TEM) images and chemical analyses of the samples were performed using a JEOL (Val-de-Reuil, France) model ARM200-CFEG microscope working at 200 kV. The EDX analyses and chemical mappings were performed using a JEOL Centurio detector.

X-ray photoelectron spectroscopy (XPS) was carried out using ESCALAB MkII (from VG Scientific, UK now part of Thermo Fisher Scientific, Waltham, MA, USA) and the processing of the measured spectra has been described in [6–8]. It is well established that a partial reduction of ceria during XPS measurements occurs as a function of irradiation time. In accordance with the literature, modification of the Ce3d spectra was observed after 30 min of X-ray exposure [36]. For this reason, the spectra were collected immediately after turning on the X-rays to avoid any X-ray induced artefacts.

Temperature-programmed reduction (TPR) and temperature-programmed oxidation experiments are described in [6].

Oxygen-TPD data were obtained by using of oxygen gas analyzer Teledyne Mod. 802 (paramagnetic principle). The samples pre-treatment consisted of heating in flow of 5% $O_2$ in $N_2$ at 450 °C for 6 h, then cooling down to room temperature in the same gaseous mixture. The applied heating rate was 7.5 K min$^{-1}$, gas flow rate: nitrogen, 500 mL min$^{-1}$.

Temperature-programmed reduction by propane ($C_3H_8$/TPR) was carried out consequently after the $O_2$-TPD by using a gas mixture of 0.125 vol.% of propane in nitrogen

(500 mL min$^{-1}$). The gas analysis was performed by on-line gas analyzers (Maihak-Sick Mod. 710S, SICK AG, Hamburg, Germany, 2012) $CO/CO_2/O_2$ and by an on-line gas analyzer for total organic content: Horiba, Model FIA-510, HORIBA Ltd, Kyoto, Japan, 2012, (equipped with flame ionization detector), mass-flow control system (Bronkhorst High-Tech B.V., Nijverheidsstraat 1A, NL-7261 AK Ruurlo, The Nederlands) being used for the gas dosing.

### 3.3. Catalytic Activity

The catalytic properties were tested in the complete oxidation of methane, propane, and *n*-hexane.

The catalytic properties were tested in the complete oxidation of methane, propane, and *n*-hexane. The catalytic tests were carried out in a fixed bed reactor with a gaseous hourly space velocity (GHSV$_{STP}$) of 60.000 h$^{-1}$. The catalysts (particle sizes of 0.3–0.6 mm) were held between plugs of quartz wool in a quartz tubular vertical flow reactor (ø = 6 mm) and the reactor diameter was 6 mm (D$_{reactor}$/D$_{particles}$ $\geq$ 10). Methane, propane, and n-hexane inlet concentrations in synthetic air were kept at 1000 (and 2000) ppm, respectively. Reactant gases were supplied through electronic mass flow controller system (Bronkhorst). On-line gas-analyzers ($CO_2/O_2$, Teledyne, Model T803, Teledyne API, Carroll Canyon Road, San Diego, CA 92131, USA, 2011) and THC- flame ionization detector) (Horiba) for analysis of total hydrocarbon content, were applied for the gas analysis.

### 4. Conclusions

The catalysts' performance of a series of mono-component cobalt and bi-component Co–Ce catalysts supported on SBA-15 in combustion of methane, propane, and *n*-hexane showed high selectivity to $H_2O$ and $CO_2$, the extent of incomplete oxidation to CO being negligible.

The results from the instrumental methods showed that the preparation of the Co–Ce catalysts by the 'two-solvent' technique does not significantly change the mesoporous structure, however, its pores are clogging with the Co and Ce guest species. Cobalt and cerium are uniformly distributed and preferentially fill up the channels of SBA-15, but oxide agglomerates located on the surface are observed as well.

The fine dispersion of cobalt and cerium oxide and their strong interaction in the channels of the SBA-15 molecular sieve, leads to the formation of difficult-to-reduce oxide phases and, consequently, to lower catalytic activity compared to monocomponent cobalt oxide catalyst. The highest catalytic activity of the mono-component cobalt sample is explained by its higher reducibility as a result of lower interaction of the cobalt oxide with the SBA-15. Despite the lower activity of the two-component Co–Ce catalysts compared to the monocomponent cobalt one, it can be supposed that they would be promising catalysts for the VOCs' oxidation due to their acceptable activity at reduced cobalt content.

The observed insignificant changes in the mesoporous structure, morphology, oxide particle size, and oxide phases after reaction prove that the mesoporous structure can prevent the agglomeration of the oxide particles, thus offering new opportunities for development of a stable catalyst for the prevention of greenhouse gas emissions.

**Supplementary Materials:** The following are available online at https://www.mdpi.com/2073-4344/11/3/366/s1, Figure S1: TEM image and EDX Co and Ce element mappings of CoCe(1:0.5) catalyst before (A,B,C) and after (D,E,F) reaction, Figure S2: TEM images of (a,b,c) and EDX Co and Ce element mappings for CoCe(1:2) catalyst before reaction, Figure S3: TEM images of (a,b,c) and EDX Co and Ce element mappings for CoCe(1:2) catalyst after reaction, Figure S4: CO formation over bi-component CoCe catalysts.

**Author Contributions:** S.T., J.-L.B., B.L., A.N. and R.V.: results analysis, manuscript preparation; conceptualization, catalytic experiments, and discussion; A.N. and R.V. catalytic test, C$_6$H$_8$-TPR, O$_2$-TPD experiments and analysis; H.K. performed and discussed XPS analysis; L.M., B.L., L.V. and J.-L.B.: BET, SAXS, XRD, TEM measurements and analysis; K.T. H$_2$-TPR measurements; A.D. and P.G.—sample synthesis; supervision, S.T., J.-L.B.; S.T., A.N. and R.V. funding acquisition. All authors

contributed to discussion of the manuscript. All authors have read and agreed to the published version of the manuscript.

**Funding:** The authors express their gratitude to the National Science Fund of Bulgaria for the financial support under the Contract КП-06-Н49/4. Research equipment of distributed research infrastructure INFRAMAT ДО1- 382/18.12.2020 (part of Bulgarian National roadmap for research infrastructures) supported by Bulgarian Ministry of Education and Science was used in this investigation.

**Data Availability Statement:** The data presented in this study are available on request from the corresponding author.

**Conflicts of Interest:** The authors declare no conflict of interest.

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
