# Peer review of "Co–Ce Oxides Supported on SBA-15 for VOCs Oxidation"

_catalysts, doi:10.3390/catal11030366_

Round 1
Reviewer 1 Report
1. The manuscript evaluates the performance of different Co-Ce oxides supported on SBA-15 for the oxidation of some model compounds of industrial VOCs. The authors have performed an intensive characterization of the catalysts before testing their performance. The work faces an interesting topic, but prior to be accepted for its publication some important modifications are required.
- The subgraphs collected in Figure 1 must be identified as Figure 1a and b. This way, the references in the text will be understand more easily. Additionally, check the legends in the subgraph corresponding to the results obtained with the catalysts after reaction, as there is a mistake on it.
- The values for the specific surface and total pore volume of the SBA-15 indicated in line 113 are not in agreement with the values collected in Table 1. Please correct it.
- The subgraphs collected in Figure 2 must be identified as Figure 2a and b. This way, the references in the text will be understand more easily.
- Table 2 must be placed in line 177 after being firstly cited in the text.
- I suggest unifying Figures 7 and 8 in a unique Figure 7 with two different subgraphs.
- Table 3 must be placed in line 237 after being firstly cited in the text.
- Figure 10 must be placed in line 274 after being firstly cited in the text.
- Figure 11 must be placed in line 289 after being firstly cited in the text.
- Why there is no data for Co-SBA-15 on Figure 12? Moreover, in the text (lines 243-244) it is said that C3H8/TPR experiments are performed up to 550 °C, but the data collected in Figure 12 did not even reach 500 °C, why?
- I suggest maintaining the same legend in Figures 13 and 14, i.e., use circles for the reaction with hexane and up triangles for the reaction s with propane.
- I did not find which is the metal content of the catalysts, just the molar ratio between Co and Ce. Did the authors measure it by ICP or XRF?
- Finally, the authors have concluded that Co-SBA-15 offers by far the best performance of all the tested catalysts. Hence, I did not see the point of preparing Co-Ce bimetallic catalysts. Which is the final goal of these bimetallic catalysts?
Reviewer 2 Report
February 24, 2021
Review – Blin, et al. ; Catalysts – “Co-Ce oxides supported on SBA-15 for VOCs oxidation”
The authors describe a study of Co oxide and Co/Ce oxide mixtures on the mesoporous silicate SBA-15 is evaluated as an oxidation catalyst for methane, propane and n-hexane. Materials are characterized by nitrogen adsorption, SAXS and XRD, TEM, and XPS. The work builds upon their prior work with Co and Mn catalysts supported on SBA-15 (Todorova et. al. Catalysis Today (2020) 357, 602-612). Overall, the work appears competently executed, thorough, and well-described. While Co3O4 catalysts have been widely studied as oxidation catalyst, the novelty of this work is the application of a ‘two-solvent’ synthesis procedure to prepare combinations of Co and Ce oxides on the SBA-15 support.
The work is publishable; however, I have concerns that should be addressed before it receives full support:
- Throughout: The authors describe their samples using the mole ratios in the aqueous precursor solutions; however, their EDX analysis reveals very different ratios of Co/Ce in the catalyst solid. For example, the material described (named) as having 1:05 Co:Ce ratio has a Co/Ce ratio of ~11 by EDX analysis. This naming convention is misleading and should be changed to accurately describe the Co/Ce ratio in the active catalyst.
- Line 124: the authors describe the reduction in surface area and pore volume from the addition of the active phase as ‘ mesopore clogging’; and in Line 170 as the mesopores are ‘filled up. This is misleading. The analysis should account for the filling of micropores, which will significantly reduce surface area and somewhat reduce total pore volume.
- Line 151-161: The EDX elemental mappings in Figures 3 – 6 are difficult to read and are not helpful. Recommend omitting or moving to Supplemental Information.
- Line 214: What table are the authors referring to in the parenthetical expression? Table 3?
- Line 220-224: This paragraph should be moved to the Methods section.
- Line 229, 235: The assignment and description of fitted curves does not appear to match the color coding in Figure 9, right image. In the written description, the peaks at ~ 882 and 884 are assigned to Ce+3, but are given different colors in the Figure.
- Line 244 (Figure 10), Lines 249 and 250: the temperatures in the Figure (355 and 480) do not match the values in the text below (356 and 481).
- Line 314: In Figure 13, switch the sample order in the figure legend to show increasing or decreasing Ce content in samples; that is, make the order Co → CoCe(1:05) →CoCe(1:2) or the reverse.
- Line 331: Figure 14 does not significantly contribute to the overall results and conclusions in this paper. Recommend either omitting or moving to Supplemental Information.
Reviewer 3 Report
"Cobalt oxide is reported as quite promising among the metal oxides used for preparation of supported catalysts for the removal of CO even at room and lower temperatures, but is more toxic element" – more toxic than what?
"The combination of Co3O4 and other different oxides such as CeO2, ZrO2, Al2O3, etc. often affects the textural, morphological, redox, acidic and basic properties, and the catalytic activity. " This is too general. It has to be stated if the changes are good or bad for specific combination.
"On the other hand, ceria can enhance the dispersion of supported metals or stabilize the oxide support toward thermal sintering" I suggest to use the word "against" instead of "toward".
"Studies have shown that the porosity of support modifies the catalytic properties by affecting the particle dispersion and reducibility of metal species." This statement needs citation.
"measurement error" mentioned (line 126) is rather measurement uncertainty
"According to literature [16, 17], Ce4+ and Ce3+ are present together, although the kind of cerium salt used as precursor and the treatment atmosphere can affect the relative intensities." Rather relative content or concentration
There is something wrong with following sentence: Huang and co-authors ascribed reduction peak around 600 °C for Co3O4–CeO2 catalysts to the reduction of Co2+ interacting with CeO2 to Co as well as the bulk oxygen of CeO2." It is hard to guess which part refers to reduction and which to the contact between phases.
"It is very likely that in our case the consumption of hydrogen over 600 °C is due on the one hand to the reduction of Co2+ (Co2+ is confirmed from XPS data) strongly interacting with cerium oxide and on the other hand to the formation of cobalt silicate during reduction due to the strong interaction of cobalt oxide with the carrier." There is no proof of formation of such silicates, it is rather a speculation and as such should be removed.
Some explanation on the efficiency of the 'bi-component' material is needed. In theory, they should be more effective than the single-component ones, but they are not.
I also miss the general conclusion – which catalyst would be preferred for VOC removal.
Round 2
Reviewer 1 Report
The authors have properly addapted the manuscript according to the comments and suggestions provided in the previous review. Thus, it can be published in Catalysts journal.